# Treatment Strategies for Non-Small-Cell Lung Cancer with Comorbid Respiratory Disease; Interstitial Pneumonia, Chronic Obstructive Pulmonary Disease, and Tuberculosis

**DOI:** 10.3390/cancers16091734

**Published:** 2024-04-29

**Authors:** Ryota Otoshi, Satoshi Ikeda, Taichi Kaneko, Shinobu Sagawa, Chieri Yamada, Kosumi Kumagai, Asami Moriuchi, Akimasa Sekine, Tomohisa Baba, Takashi Ogura

**Affiliations:** Department of Respiratory Medicine, Kanagawa Cardiovascular and Respiratory Center, 6-16-1, Tomioka-higashi, Kanazawa-ku, Yokohama 236-0051, Japan; ootoshi.1540m@kanagawa-pho.jp (R.O.); kaneko.9150n@kanagawa-pho.jp (T.K.); shinobu.sagawa1012@gmail.com (S.S.); yamada.9040n@kanagawa-pho.jp (C.Y.); kumagai.8y20n@kanagawa-pho.jp (K.K.); moriuchi.cr50o@kanagawa-pho.jp (A.M.); sekine.1v50c@kanagawa-pho.jp (A.S.); baba.19049@kanagawa-pho.jp (T.B.); ogura.0302b@kanagawa-pho.jp (T.O.)

**Keywords:** acute exacerbation, chronic obstructive pulmonary disease, cytotoxic anti-cancer drug, immune checkpoint inhibitor, interstitial pneumonia, lung cancer, non-small cell lung carcinoma, tuberculosis

## Abstract

**Simple Summary:**

Lung cancer patients are frequently complicated by various respiratory diseases during their course. In particular, interstitial pneumonia and chronic obstructive pulmonary disease are often associated with lung cancer due to their common pathogenesis, and acute exacerbations of these diseases can be fatal. Therefore, it is important to select a therapy that is less likely to induce acute exacerbations of interstitial pneumonia and chronic obstructive pulmonary disease. Furthermore, lung cancer patients are at high risk of developing or reactivating pulmonary tuberculosis triggered by pharmacotherapy and often struggle with the diagnosis and treatment of tuberculosis complicated by lung cancer. This review summarizes the current evidence regarding pharmacotherapy for lung cancer patients with interstitial pneumonia, chronic obstructive pulmonary disease, and pulmonary tuberculosis and discusses future prospects.

**Abstract:**

Non-small cell lung cancer (NSCLC) patients are often complicated by other respiratory diseases, including interstitial pneumonia (IP), chronic obstructive pulmonary disease (COPD), and pulmonary tuberculosis (TB), and the management of which can be problematic. NSCLC patients with IP sometimes develop fatal acute exacerbation induced by pharmacotherapy, and the establishment of a safe treatment strategy is desirable. For advanced NSCLC with IP, carboplatin plus nanoparticle albumin-bound paclitaxel is a relatively safe and effective first-line treatment option. Although the safety of immune checkpoint inhibitors (ICIs) for these populations remains controversial, ICIs have the potential to provide long-term survival. The severity of COPD is an important prognostic factor in NSCLC patients. Although COPD complications do not necessarily limit treatment options, it is important to select drugs with fewer side effects on the heart and blood vessels as well as the lungs. Active TB is complicated by 2–5% of NSCLC cases during their disease course. Since pharmacotherapy, especially ICIs, reportedly induces the development of TB, the possibility of developing TB should always be kept in mind during NSCLC treatment. To date, there is no coherent review article on NSCLC with these pulmonary complications. This review article summarizes the current evidence and discusses future prospects for treatment strategies for NSCLC patients complicated with IP, severe COPD, and TB.

## 1. Interstitial Pneumonia

### 1.1. Introduction

Approximately 5–15% of lung cancer (LC) patients have interstitial pneumonia (IP) at the time of diagnosis [1,2,3]. Contrarily, the incidence of LC in patients with IP is reported to be 10–20%, which is 5–14 times higher than the incidence in the general population [1,4]. The common risk factors for both IP and LC include smoking, bacterial and viral infections, environmental and occupational exposures, and chronic tissue damage [5,6]. Additionally, the common molecular mechanisms include genetic and epigenetic mutations, abnormal micro-RNA expression, inhibition of apoptosis associated with the activation of intracellular signal transduction, and weakening of cellular interactions [7,8]. These common etiologies result in a high rate of LC and IP complications.

Several studies have reported that LC patients with IP have a worse prognosis than patients with IP alone or LC alone. In the Hokkaido study, a large observational study involving 553 idiopathic pulmonary fibrosis (IPF) patients in Japan, LC was the third leading cause of death (11%) [9]. In an observational study involving 181 IPF patients, Tomassetti et al. reported that the median survival of IPF patients with LC was significantly shorter than that of IPF patients without LC (38.7 vs. 63.9 months, *p* < 0.001) [10]. Additionally, in a retrospective study of 637 LC patients to determine the impact of IPF on prognosis, 34 (5.3%) LC patients with IPF had a worse prognosis than those without IPF, regardless of the stage or treatment received [11].

Despite the high complication rate and poor prognosis of LC patients with IP, almost all clinical trials for LC exclude this population because of concerns about chemotherapy-induced acute exacerbations of IP. Acute exacerbation is a fatal complication in IP patients, with a mortality rate of 30–50% [12,13]. Patients with IPF develop acute exacerbations at a frequency of 10–15% per year during the natural course of the disease, and patients with IP other than IPF, including non-specific interstitial pneumonia and IP associated with collagen vascular disease, also develop acute exacerbations at a frequency of 3–5% per year [14,15,16]. Furthermore, the incidence of chemotherapy-induced acute exacerbations of IP has been reported to be 5–20%; thus, it is most important to select a treatment that is less likely to induce acute exacerbations of pre-existing IP [17,18,19].

Recently, LC treatment has made significant advances, and newer therapies such as carbon-ion radiotherapy and high-power microwaves are expected to be relatively safe treatment options for LC patients with IP complications [20,21]. However, most LC patients are already advanced at diagnosis, and pharmacotherapy remains the cornerstone of treatment. In particular, clinicians are often faced with the difficult choice of pharmacotherapy for patients with non-small cell lung carcinoma (NSCLC), which has various treatment options, including molecular-targeted drugs and immune checkpoint inhibitors (ICIs), in addition to cytotoxic chemotherapy. In the following sections, we summarize the current evidence and discuss future prospects for pharmacotherapy for advanced NSCLC with comorbid IP.

### 1.2. Risk of Acute Exacerbation by Pharmacotherapy for NSCLC with Comorbid IP

Based on previous reports, pharmacotherapy for NSCLC patients with comorbid IP induces acute exacerbation of IP with a frequency of approximately 5–20% [17,18,19]. The exact mechanism by which pharmacotherapy causes acute exacerbations of IP is not known, but direct cell damage by reactive oxygen species and proteolytic enzymes and activation of immune cells have been proposed [22]. The risk factors for the development of acute exacerbations of IP due to cytotoxic chemotherapy have been reported in several studies. The patients with the usual interstitial pneumonia (UIP) pattern on computed tomography (CT) reportedly have a higher frequency of acute exacerbations induced by cytotoxic chemotherapy than those with a non-UIP pattern (30% vs. 8%, *p* = 0.005) [23]. Additionally, low forced vital capacity (FVC) is also reported to be associated with a higher risk of acute exacerbation induced by cytotoxic chemotherapy, and low FVC is suggested to be more strongly associated with the risk of acute exacerbation than UIP pattern on CT [24]. Additionally, a modified GAP index, a scoring system to evaluate IPF severity, is also reported to be associated with a higher incidence of chemotherapy-induced acute exacerbations and a lower 1-year survival rate [25,26]. Although there are no established risk factors or biomarkers that can actually predict chemotherapy-induced acute exacerbation, these reports suggest that the administration of chemotherapy for NSCLC patients with IP should be considered based on a comprehensive assessment of the patient’s age, general condition, IP severity, and LC prognosis, with careful consideration of the risk–benefit ratio.

In the pharmacotherapy for NSCLC with comorbid IP, it is important to identify and select drugs that are unlikely to cause acute exacerbations of IP. The Japanese Respiratory Society issued a “Statement” on the treatment of LC with comorbid IP in 2017 [27]. This statement classified cytotoxic chemotherapy into three categories according to the risk of acute exacerbation of IP, and several chemotherapy agents, including platinum-containing drugs, etoposide, paclitaxel, and vinorelbine, are classified as drugs that can be administered with caution. In actual practice, first-line treatment may be considered from among these treatment options reported to be relatively safe. However, there is no evidence of efficacy beyond the second-line treatment, and it is not recommended at this time.

ICIs are often reported to carry a high risk of causing acute exacerbation of pre-existing IP [28,29]. A Japanese retrospective study of NSCLC patients treated with ICIs showed a higher incidence of pneumonitis in patients with IP than in those without IP (29% vs. 10%, *p* = 0.027) [30]. Furthermore, a recent meta-analysis of chemotherapy, including ICIs for LC with comorbid IP, showed that ICI-associated pneumonitis occurred at a high rate of 27–30% for all grades and 12–15% for grade 3 or higher [31,32]. Based on these results, ICIs are generally considered to be at a high risk for acute exacerbation of IP. However, in these reports, whether there are differences in the risk for developing acute exacerbations due to the pre-existing IP subtypes, the specific radiological findings, or pulmonary function has not been fully investigated. In pilot (*N* = 6) and phase II (*N* = 18) trials of nivolumab in previously treated NSCLC patients with mild IP defined as (1) no honeycomb lung, (2) negative autoantibody, and (3) %VC ≧ 80% (Commonly called “HAV Criteria”, an acronym for Honeycomb lung, autoantibody, and VC), the frequency of drug-induced pneumonitis ranged from 0–11% [33,34]. On the other hand, in a phase II trial of atezolizumab in NSCLC patients with chronic fibrotic IP, with %FVC > 70%, with or without honeycomb lung, grade 3 or higher pneumonitis occurred at a rate of 24% [35]. The post-hoc analysis of this study suggested that the presence of honeycomb lung may be a risk factor for pneumonitis, although it is not statistically significant. Meanwhile, the aforementioned meta-analysis of ICIs for NSCLC patients with IP, which included three interventional studies mentioned above and seven retrospective studies, reported that honeycomb lung may not be a risk factor for pneumonitis [33]. Based on these results, the safety of ICIs for NSCLC patients with IP has not yet been established, and currently, there is no rationale for its use in first-line therapy. The risk factors for acute exacerbations of IP caused by ICIs are still unknown, and further accumulation of a large number of data and a detailed analysis of the risk factors are required in the future.

For NSCLC with driver gene mutations/translocations, including epidermal growth factor receptor (EGFR), anaplastic lymphoma kinase, and c-ros oncogene 1 (ROS1) genes, the first-line therapy with tyrosine kinase inhibitors (TKIs) targeting gene mutation/translocation is recommended. However, when gefitinib-induced pneumonitis occurred frequently in Japan, pre-existing IP was identified as an independent risk factor for pneumonitis [17,36]. Since then, special caution has been required when administering molecular-targeted drugs to driver gene mutation-positive NSCLC patients with comorbid IP. In fact, it has been reported that only 0.4% of lung adenocarcinoma patients with EGFR mutations have IP; thus, there are few situations in real-world clinical practice where we wonder whether EGFR-TKIs should be administered to these populations [37]. Contrarily, with regard to gene mutations, such as KRAS and BRAF, which are relatively common among smokers, patients may potentially have a higher frequency of IP complications. In fact, it has been suggested that NSCLC patients with IP may have more KRAS and BRAF mutations than those without IP [38]. A previous case study reported the KRAS inhibitor sotrasib being administered safely and effectively to KRAS G12C positive NSCLC patients with comorbid IP [39]. There are no detailed data on the prevalence of IP or risk factors for acute exacerbation of IP in NSCLC patients with driver mutations/translocations other than EGFR, and further studies are still needed.

### 1.3. First-Line Treatment of NSCLC with Comorbid IP

Table 1 summarizes the prospective clinical trials reported to date on the first-line treatment of advanced NSCLC with comorbid IP.

First, carboplatin plus nanoparticle albumin-bound paclitaxel (nab-paclitaxel) may be considered as first-line chemotherapy for advanced NSCLC with comorbid IP. Until now, two single-arm phase II trials of carboplatin plus nab-paclitaxel in advanced NSCLC with comorbid IP, which were multicenter prospective studies involving a relatively large number of patients (94 and 36 patients, respectively), have been reported [40,41]. Both studies showed consistent safety and efficacy results with an incidence of acute exacerbation of existing IP of 4.3–5.6%, response rates of 51–56%, median progression-free survival (PFS) of 5.3–6.2 months, and median overall survival (OS) of 15.1–15.4 months. Based on these results, this regimen may be considered as standard first-line treatment for NSCLC comorbid with IP.

Two prospective trials also investigated the combination therapy of carboplatin plus weekly paclitaxel for advanced NSCLC with comorbid IP [18,42]. Although these trials involved fewer patients than the aforementioned trials (35 and 18 patients, respectively), the incidence of acute exacerbation of IP induced by this combination therapy has been reported at 5.6–12.1%, the response rate of 61–70%, median PFS of 5.3–6.3 months, and median OS of 10.6–19.8 months suggesting that this may also be a promising treatment option.

Additionally, two prospective trials of carboplatin plus S-1 in 21 and 33 patients, respectively, reported acute exacerbation rates of 6.1–9.5%, response rates of 33.0–33.3%, median PFS of 4.2–4.8 months, and median OS of 9.7–12.8 months [43,44]. Based on these results, carboplatin plus S-1 therapy may be another option for the first-line treatment for NSCLC comorbid with IP. However, instead of using S-1 in the first-line treatment, preservation of S-1 for the second-line treatment, described below, may also be considered.

The addition of angiogenesis inhibitors, including bevacizumab or ramucirumab, for NSCLC patients comorbid with IP, is controversial but may be relatively safe. In a retrospective study of 51 NSCLC patients with IP, those treated with bevacizumab had better rates of acute exacerbation of IP (0% vs. 22.6%) and median PFS (8.0 vs. 4.3 months) than those treated without bevacizumab [47]. Similarly, several retrospective studies have shown that the combination of bevacizumab was relatively safe to use in NSCLC patients with IP [48,49]. Recently, a multicenter prospective phase II study of carboplatin plus weekly paclitaxel plus bevacizumab for advanced NSCLC with comorbid IP was conducted in Japan [45]. Although this was a single-arm study of 17 patients, this study showed relatively safe and efficacy results with an incidence of acute exacerbation of existing IP of 5.9%, response rates of 52.9%, median PFS of 5.7 months, and median OS of 12.9 months. From these results, the concomitant use of bevacizumab is unlikely to change the risk of acute exacerbations of IP and may be considered in patients who are eligible for administration.

Recently, monotherapy with ICIs or combination therapy with ICIs and cytotoxic chemotherapy has become the standard of care for the first-line treatment of advanced NSCLC. However, as noted above, many studies have shown that pre-existing IP increases the risk of ICI-associated pneumonitis, and most ICI package inserts state that they should be administered with caution to LC patients with comorbid IP [30,31,32]. Therefore, ICIs should not be used as first-line treatment for NSCLC patients with IP at this time.

### 1.4. Second-Line Treatment of NSCLC with Comorbid IP

To date, no prospective interventional study of cytotoxic chemotherapy as second or subsequent-line therapy for NSCLC with comorbid IP has been reported. In a nationwide survey on second-line therapy of 278 LC patients with IP in Japan, the acute exacerbation rate of IP was reported for each cytotoxic anti-cancer drug, with a frequency of 15.3% for docetaxel and 28.6% for pemetrexed [50]. Since other retrospective studies have also reported a relatively high rate of acute exacerbations of IP with these agents, docetaxel and pemetrexed should be recognized as relatively high-risk drugs for NSCLC patients with comorbid IP [51,52,53]. S-1 monotherapy is thought to be relatively less likely to induce acute exacerbations of IP based on the results of retrospective studies. According to the aforementioned national survey for LC patients with comorbid IP, no patient experienced an acute exacerbation of pre-existing IP when treated with S-1 alone [50]. Additionally, as mentioned above, the low incidence of acute exacerbations (6.1–9.5%) was reported in the two prospective studies of carboplatin plus S-1 as first-line treatment [43,44]. On the other hand, another retrospective study reported that even S-1 caused acute exacerbation of IP in 21% of cases (3 of 14 patients) [23]. From these results, S-1 for NSCLC with comorbid IP may be less risky than second-line therapy with other cytotoxic agents, but caution should, of course, be exercised when administered.

The safety of ICIs for NSCLC patients with IP has not yet been established, but it has been suggested that ICIs may be highly effective for this population. In a pilot study of 6 NSCLC patients with mild IP that met the so-called “HAV criteria” (without honeycomb lung, without autoantibody, and %VC ≥ 80%), nivolumab monotherapy did not induce acute exacerbation of IP [33]. Furthermore, in a phase II study of 18 NSCLC patients with mild IP selected from four centers using the same criteria, nivolumab-induced pneumonitis in two patients (11%), all of whom were grade 2 and improved rapidly with corticosteroid therapy [34]. Importantly, these trials reported a high efficacy of nivolumab monotherapy in NSCLC patients with IP, with response rates of 39–50% and disease control rates of 72–100%. Similarly, a multicenter, retrospective study of 216 patients with NSCLC reported that IP patients tended to have a higher response rate than those without IP (27% versus 13%, *p* = 0.078) [54]. Because the development of IP is often associated with smoking and microsatellite instability, it has been suggested that NSCLC with comorbid IP with a high tumor mutation burden may benefit more from ICIs [7]. On the other hand, however, a phase II trial of atezolizumab in NSCLC patients with IP terminated early after the enrollment of 17 patients due to a high incidence of pneumonitis with 24% for grade 3 or higher, and 6% for grade 5 [35]. In this study, logistic regression analysis suggested that the presence of honeycomb lungs may be associated with the development of ICI-induced pneumonitis, although the results were not statistically significant.

Based on these results, S-1 monotherapy is often administered as standard second-line treatment of NSCLC with comorbid IP. However, retrospective studies of cytotoxic chemotherapy as a second-line treatment for NSCLC with IP have shown limited efficacy with 1-year survival rates of approximately 10% and little promise for long-term survival [52,53]. Therefore, for NSCLC patients with IP, who have few treatment options and a poor prognosis, ICIs have the potential to be the only existing therapy with long-term survival. Large studies identifying the risk factors for ICI-induced pneumonitis are needed for appropriate patient selection.

### 1.5. Antifibrotic Agents for NSCLC with Comorbid IP

Recently, antifibrotic agents such as pirfenidone and nintedanib have been widely used clinically worldwide for the treatment of IPF. Antifibrotic agents have been reported to slow the progression of IP and reduce the incidence of acute exacerbations of IP. In a randomized phase III trial in IPF, nintedanib significantly prevented acute exacerbations of IP during the course of the disease (1.9% and 4.7%, *p* = 0.010), with a hazard ratio of 0.53 for the time to the first acute exacerbation [55]. Furthermore, in a multicenter phase II trial of perioperative treatment of NSCLC with IPF, pirfenidone reduced the incidence of postoperative acute exacerbations of IPF [56].

Based on these results, the J-SONIC trial, a randomized phase III trial of carboplatin plus nab-paclitaxel with or without nintedanib for NSCLC patients with IPF, was conducted in Japan [46]. However, this study failed to demonstrate a reduction in the incidence of acute exacerbations of IP with concomitant nintedanib, and at this time, there is no evidence to aggressively recommend concomitant antifibrotic therapy for advanced NSCLC patients with comorbid IP.

## 2. Chronic Obstructive Pulmonary Disease

### 2.1. Introduction

Chronic obstructive pulmonary disease (COPD) and LC, both of which are mainly caused by smoking, are often combined, and the prevalence of COPD in LC patients is approximately 40–70% [57,58]. The impact of COPD on the survival of LC patients varies from report to report, but a large meta-analysis of the United States (U.S.), European, and Asian studies reported that LC patients with COPD have a poorer prognosis than those without COPD (HR, 1.17; 95% CI: 1.10–1.25) [59]. Other reports have also generally recognized COPD as a poor prognostic factor for LC patients [60,61,62]. Furthermore, the severity of COPD is considered an important prognostic factor in LC patients, and it has been reported that the more severe the Global Initiative for Chronic Obstructive Lung Disease classification grade of COPD severity, the worse the prognosis of LC patients [63]. No molecular pathway linking COPD severity and LC prognosis has been identified, and elucidation of this molecular pathway may contribute to treatment selection for LC patients with severe COPD. LC patients with severe COPD have not only low pulmonary function but also a high frequency of systemic complications, including cancer cachexia, heart failure, and diabetes mellitus, and many patients have poor performance status (PS). Thus, the decision to use pharmacotherapy for LC patients with severe COPD should be carefully considered.

In several large retrospective studies, the presence of mild-to-moderate COPD with a preserved percent-predicted forced expiratory volume in one second (%FEV1) of >50% did not significantly affect the prognosis of LC or the side effects of treatment [58,64]. Patients with severe COPD requiring home oxygen therapy have been excluded from many clinical trials and studies, and there are few reports on chemotherapy for LC. However, a retrospective observational study has reported that even advanced NSCLC patients with severe to most severe COPD had a longer OS with chemotherapy than with supportive care alone (14.0 vs. 8.0 months, *p* = 0.003) [65]. The Japanese retrospective study evaluating the efficacy and safety of chemotherapy in 40 patients with advanced LC complicated by chronic respiratory failure requiring home oxygen therapy also showed that the only factor significantly associated with improved prognosis was the use of first-line or second-line treatment (HR, 0.42; 95% CI: 0.18–0.94) [66]. Therefore, even in the cases of low pulmonary function requiring home oxygen therapy, up to the first-line or second-line treatment may be considered if the PS is maintained.

In the following sections, we summarize the current evidence and discuss future issues regarding pharmacotherapy for advanced NSCLC with severe COPD.

### 2.2. Cytotoxic Chemotherapy for NSCLC with Severe COPD

There is no clear evidence for the efficacy and safety of cytotoxic chemotherapy for severe COPD requiring home oxygenation, and no drugs are listed as contraindicated for severe COPD in the Japanese drug package insert. However, these patients with low pulmonary function can be fatal even with mild infectious pneumonia and drug-induced pneumonitis. LC patients with severe or most severe COPD (%FEV1 < 50%) reported a significantly higher rate of pulmonary adverse events such as pneumonia (46.4% vs. 31.2%, *p* < 0.001) and COPD exacerbations (30.4% vs. 6.9%, *p* < 0.001) during LC treatment than those with mild-to-moderate COPD (%FEV1 > 50%) [67]. Therefore, it is important to select cytotoxic chemotherapy with a relatively low risk of pulmonary adverse events in LC patients with severe COPD.

Additionally, COPD patients are often complicated by other comorbidities, including cardiovascular disease and diabetes, which may affect patient survival [68]. Especially in patients with severe COPD, cytotoxic chemotherapy that affects the cardiovascular system should be avoided because of the risk of worsening respiratory status even with increased cardiac load due to hypertension or massive infusion. In light of these points, carboplatin should be chosen for platinum doublet therapy because cisplatin may aggravate heart failure due to the massive infusions. Similarly, docetaxel, often used in the chemotherapy of NSCLC, carries the risk of increased cardiac load due to fluid retention and edema. Furthermore, a retrospective study involving 392 NSCLC patients reported that LC patients with emphysema are 4.95 times more likely to develop pneumonitis induced by docetaxel than patients without emphysema [69]. For these reasons, the use of docetaxel in NSCLC patients with severe COPD should be cautious.

COPD affects the expression of vascular endothelial growth factor (VEGF) in the circulation, suggesting that the anti-VEGF and anti-VEGF receptor inhibitors may be effective in NSCLC patients with COPD. A small retrospective study of NSCLC patients treated with carboplatin plus paclitaxel plus bevacizumab reported significantly longer PFS in patients with COPD (35 patients) than in those without COPD (39 patients) (6.1 vs. 3.3 months, *p* = 0.049) [70]. Thus, anti-VEGF and anti-VEGF receptor inhibitors should be considered for NSCLC patients with mild-to-moderate COPD. However, given the risk of increased cardiac burden due to fluid retention and edema, which are typical adverse events of anti-VEGF and anti-VEGF receptor inhibitors, caution should be exercised when administering these drugs to NSCLC patients with severe to most severe COPD.

In view of the above, in clinical practice, cytotoxic chemotherapy, such as pemetrexed, paclitaxel, nab-PTX, and S1 as single agents or in combination with carboplatin, is often empirically selected for NSCLC with severe COPD. To date, however, there have been no prospective trials of cytotoxic chemotherapy for this population, and the optimal treatment of choice is unknown. To resolve these clinical questions, it is necessary to accumulate data from large-scale studies in the future.

### 2.3. Immune Checkpoint Inhibitors for NSCLC with Severe COPD

Recently, it has been suggested that in patients with NSCLC, COPD comorbidity may increase the efficacy of ICIs [71,72]. A retrospective cohort study of 125 NSCLC patients treated with ICIs reported that COPD patients and current smokers had longer PFS [73]. Additionally, a subset analysis of this study reported significantly longer survival in ex-smokers with COPD complications than in ex-smokers without COPD (OS; 359 vs. 145 days, *p* = 0.035). COPD patients reportedly have altered Th1/Th2 ratios and Treg/Th17 balance due to increased Th1 and Th2 cells and increased expression of PD-1 on CD8+ T cells and PD-L1 on macrophages, which may increase the efficacy of ICIs [73,74].

On the other hand, it has been reported that a complication of COPD increases the risk of developing ICI-induced pneumonitis. The U.S. Food and Drug Administration (FDA) report on pembrolizumab, an anti-PD-1 antibody drug, found that patients with COPD had a higher incidence of pneumonitis than those without COPD (5.4% vs. 3.1%) [75]. Similarly, a recent multicenter retrospective study of ICI-related pneumonitis from the U.S. also reported that the presence of COPD increased the risk of developing pneumonitis by 2.79 times [76]. Furthermore, in a retrospective study involving 99 NSCLC patients with COPD treated with ICIs, a higher incidence of immune-related adverse events and poorer prognosis were observed in patients with more severe COPD [77]. Thus, ICI administration should be cautious in patients with severe COPD, who are at a high risk of developing ICI-related pneumonitis, which can be fatal due to the low respiratory reserve.

Additionally, COPD patients with a history of heavy smoking are often complicated by IP, and they are so-called combined pulmonary fibrosis and emphysema. As mentioned above, IP is another risk factor for the development of ICI-related pneumonitis. The stronger the emphysematous changes, the more difficult it is to determine whether IP is complicating the disease. If it is difficult to determine the presence or absence of IP complications, fine crackles on auscultation may help in making a decision [78].

### 2.4. Treatment of COPD for NSCLC with COPD

When treating LC patients with COPD, clinicians often focus solely on LC treatment and neglect the treatment of the COPD. Although there is no clear indication yet that the medical management of COPD is necessary for LC patients, many studies have shown that COPD affects the prognosis and response to chemotherapy of LC. In a small Japanese retrospective study, among the NSCLC patients with COPD, those treated for COPD with a long-acting muscarinic antagonist and/or long-acting β2 agonist (*N* = 37) had significantly longer survival than those not treated (*N* = 66) (16.7 vs. 8.2 months, *p* = 0.023) [79]. Surprisingly, approximately two out of three NSCLC patients with COPD in this study were not receiving treatment for COPD. This study also reported a positive impact of COPD treatment on prognosis in a multivariate analysis (HR, 0.52; 95% CI: 0.31–0.87), suggesting that it may be important to treat COPD as well as LC.

## 3. Pulmonary Tuberculosis

### 3.1. Introduction

Patients with tuberculosis (TB) have been reported to have a higher risk of developing LC than healthy individuals. A population-based cohort study in Taiwan showed that a history of TB increased the risk of developing LC by 1.76 times [80]. A recent large-scale prospective cohort study analyzing national medical examination data in Korea also found that the presence of TB increases the incidence of LC (HR: 1.49 for men and 1.37 for women) [81]. Conversely, patients with malignant diseases have also reported a high risk of developing active TB during their course. A large meta-analysis in the U.S. reported a particularly high risk of developing TB in hematologic cancers, head and neck cancers, and LC (incidence rate ratios = 26, 16, and 9, respectively) [82]. An observational study of 904 LC patients in Japan reported a cumulative incidence of TB of 1.38%, which is 25 times higher than that of the general population [83]. To summarize the previous Japanese reports on LC and TB, active TB is noted in 2–5% of LC cases, whereas LC is noted in 1–2% of active TB cases [84,85,86,87,88].

The mechanism of TB complicated with LC has been postulated to include proliferative changes in the columnar epithelium of the bronchial wall and squamous cell transformation due to chronic inflammation and oxidative stress of TB, as well as immune evasion of tumor cells due to the changes in cellular immune function caused by TB infection [89,90,91].

When a patient with LC is complicated with TB, there are several clinical problems. First, when a new shadow due to TB appears during the disease course of an LC patient, diagnosing TB is difficult because of the wide variety of differential diagnoses. Second, when treating TB in patients with LC, it is difficult to determine whether treatment for LC should be continued or discontinued. Additionally, the interactions between drugs for TB and LC need to be considered. In the following section, we summarize the evidence to date on how to manage when LC patients develop TB.

### 3.2. Diagnosis of TB Complicated by LC Patients

As noted above, LC patients with reduced cellular immunity are at a high risk of developing active TB. In addition to known risk factors, including older age, male sex, history of TB, and gastrectomy, treatment for LC, including cytotoxic chemotherapy and ICIs, increases the risk of developing TB [92,93,94]. Notably, a series of cases of TB developing during the course of ICI treatment have recently been reported [95,96,97,98]. In a Japanese retrospective study involving 297 LC patients treated with ICIs, five patients (1.7%) developed TB during the treatment course [99]. Importantly, three out of the five patients had pulmonary TB, whereas the other two had extrapulmonary TB (cervical and hilar lymph nodes and knee joints), suggesting that TB development during the ICI treatment requires attention not only to the lungs but also to the whole body. Interestingly, in a study analyzing the FDA database, the use of anti-PD-1 or PD-L1 antibodies was associated with the development of TB, whereas the use of anti-CTLA-4 antibodies did not increase the risk of developing TB [100]. The etiology of TB induced by ICI treatment is speculated to be a mechanism by which *Mycobacterium tuberculosis* evades the host immune response by inhibiting PD-1, which induces suppression of IFN-γ production and over-secretion of TFN-α [90,98]. Therefore, when treating LC patients with pharmacotherapy, especially anti-PD-1/PD-L1 antibodies, the risk of developing TB should always be kept in mind.

When LC and TB are combined, the diagnosis of either disease is often delayed because of the difficulty in differentiating the two diseases from each other, and this delay worsens the patient’s prognosis [101,102,103]. Therefore, it is important to detect and diagnose TB in LC patients as early as possible. Despite the good response to systemic chemotherapy in LC patients, the emergence of isolated sites or new foci, especially in typical anatomical sites of TB infection (e.g., upper lobes or apical lower lobes of the lung), should draw attention to the possibility of complicating a TB infection [104]. A previous retrospective study comparing LC patients with and without TB found that LC patients with TB were more likely to have symptoms such as cough, hemoptysis, nocturnal sweating, and more diverse CT findings such as lobar signs of mass, calcified lesions, pleural thickening, and hemorrhagic pleural effusion [105,106]. If these findings are present in LC patients, the possibility of developing TB complications should be considered.

Several meta-analyses and position statements have recommended that the IFN-γ releasing assay (IGRA) be performed as a screening procedure prior to the initiation of systemic chemotherapy, especially ICIs [82,107,108]. On the other hand, patients with malignant tumors were more likely to have false-negative results and weaker responses to IGRA due to their decreased immunocompetence [109,110,111]. Thus, although the usefulness of IGRA as a screening tool prior to chemotherapy for LC is still controversial, the IGRA measurements may be considered prior to ICIs in LC patients with lung lesions suggestive of chronic infection or at risk for the aforementioned TB infection [96].

### 3.3. Treatment Strategy for LC Patients with TB

Reports on the efficacy and safety of therapy for LC during TB treatment are limited, and there is no clear evidence yet. However, several reports have shown that cancer treatment does not interfere with TB treatment. A retrospective study conducted in Japan involving 30 patients with malignant tumors complicated by TB (including 15 LC patients) reported that concurrent administration of anti-TB drugs and cytotoxic chemotherapy was effective and safe [103]. Similarly, in a Korean retrospective case–control study, TB that developed during chemotherapy was not clinically different from TB that developed under normal circumstances, and chemotherapy did not interfere with TB treatment [92]. Recently, it has been reported that the negative conversion rate of sputum culture with standard TB treatment after 2 months was good at 94%, even in LC patients with TB; furthermore, no significant difference was observed in the outcome of LC treatment between the LC patients with and without TB complications [94,112].

There are no clear criteria for when to discontinue or resume lung cancer treatment in LC patients with active TB. The risk of infecting medical staff with TB must also be considered, and ideally, it may be desirable to precede TB treatment if LC treatment, such as chemotherapy, can be postponed. Because the duration of isolation hospitalization for TB is approximately 2 months, it is sometimes possible to treat LC without missing the treatment opportunities even after waiting for the negative conversion of sputum with TB treatment. However, the priority given to TB treatment should not result in missed opportunities for radical LC treatment, such as surgery or radiation chemotherapy. A previous Japanese retrospective study of 24 patients with TB-associated malignancies indicated that resumption of cancer treatment 2 months after completion of TB treatment may worsen the prognosis due to cancer progression [113]. Several papers have so far made suggestions for the timing of LC treatment for patients with concomitant TB. Ho et al. recommend that surgery and cytotoxic chemotherapy be administered 2–3 weeks after starting TB treatment and that molecularly targeted drugs be started while assessing drug interactions with anti-TB drugs and liver function [104]. The Japanese Society of Tuberculosis suggests that surgery should be performed 4 weeks after the start of TB treatment, and after confirmation of a negative smear, radiotherapy can be started at the same time as the TB treatment, and chemotherapy should be started 2–3 weeks after starting the TB treatment [114]. In our opinion, we offer the following treatment suggestions based on the previous reports [92,103,104,113,114] (Figure 1).

### 3.4. Interactions between Drugs for Tuberculosis and Lung Cancer

When anti-Tb drugs and anti-cancer drugs are given simultaneously, attention should be paid to drug interactions between them. Among the anti-TB drugs, rifampicin is a potent inducer of cytochrome P450, which may attenuate the effect of some anti-cancer drugs [115]. In particular, rifampicin reduces the area under the concentration–time curve of most molecular-targeted drugs by 60–80%, and caution should be exercised when using them concomitantly [103,116]. The interactions between the molecular-targeted drugs and rifampicin, based on the drug’s package insert, are summarized in Table 2.

Additionally, among cytotoxic anti-cancer drugs, etoposide, irinotecan, paclitaxel, docetaxel, and vinorelbine are metabolized by CYP3A4 and may be less effective when combined with RFP [103,117]. Therefore, during TB treatment, including rifampicin, the use of these antitumor drugs should be avoided as much as possible if alternative drugs are available. If the use of these antitumor drugs is absolutely necessary, switching to rifabutin, which has a weaker CYP induction effect, instead of rifampicin, should be considered as an option.

## 4. Conclusions

LC patients are often complicated by other respiratory diseases, with the management of which being problematic. This review article summarizes the current evidence and discusses future prospects for treatment strategies for LC patients complicated by IP, severe COPD, and TB.

Approximately 5–15% of LC patients have IP, and it is most important to select a treatment that is less likely to induce acute exacerbations of pre-existing IP. For first-line treatment of advanced NSCLC with comorbid IP, carboplatin plus nab-PTX is the treatment regimen with the most reported efficacy and safety. Although the safety of ICIs for NSCLC with IP is still controversial, it has the potential to be the only existing therapy with long-term survival.

The prevalence of COPD in LC patients is approximately 40–70%; the severity of COPD is considered an important prognostic factor in LC patients. However, even in the cases of low pulmonary function requiring home oxygen therapy, up to the first-line or second-line treatment may be considered. In pharmacotherapy for NSCLC with severe COPD, it is important to select agents that cause fewer pulmonary adverse events and cardiovascular burden. In addition, clinicians should not forget to treat COPD.

Active TB is noted in 2–5% of LC cases, whereas LC is noted in 1–2% of active TB cases. When treating LC patients with pharmacotherapy, especially ICIs, the risk of developing TB should always be kept in mind. Ideally, it may be desirable to precede TB treatment until the risk of TB infection has decreased, but this should not result in the loss of the opportunity for radical treatment of LC. In the treatment of LC patients with active TB, consider adding LC treatment after 2–3 weeks of prior TB treatment.

This review article summarizes the current evidence and discusses future prospects for treatment strategies for LC patients complicated by IP, severe COPD, and TB. However, large prospective studies on LC patients complicating these respiratory diseases are limited, and the evidence is insufficient. In particular, although the safety of ICIs for NSCLC with these complications is still controversial, it has the potential for long-term survival. It is particularly important to identify risk factors and biomarkers that predict exacerbation of pre-existing lung disease by ICIs. For appropriate patient selection, large studies are warranted in the future to identify the risk factors for ICIs in NSCLC patients with these complications.

## Figures and Tables

**Figure 1 cancers-16-01734-f001:**
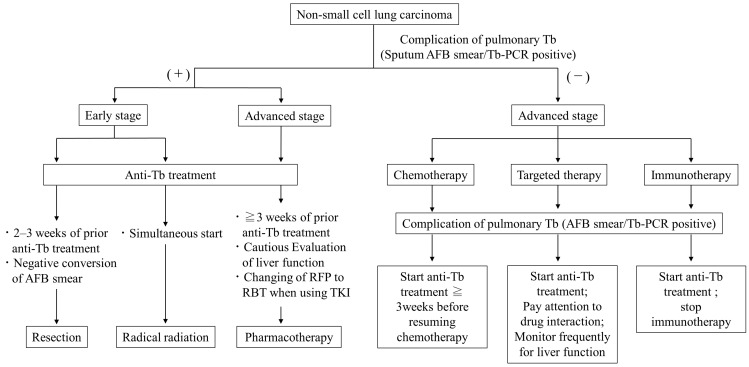
Suggested algorithm for timing of lung cancer treatment in patients with comorbid sputum-smear/PCR positive active pulmonary tuberculosis.

**Table 1 cancers-16-01734-t001:** Major prospective study on non-small cell lung carcinoma complicated by interstitial pneumonia.

Line	Phase	Study Design	Treatment Regimen	Number	PFS	OS	ORR	Incidence of AE of IP	Reference
First-line	2	Single arm	CBDCA + nab-PTX	94	6.2	15.4	51%	4.3%	[40]
First-line	2	Single arm	CBDCA + nab-PTX	36	5.3	25.4	55.6%	5.6%	[41]
First-line	Pilot	Single arm	CBDCA + weekly PTX	18		10.6	61%	5.6%	[18]
First-line	2	Single arm	CBDCA + weekly PTX	35	6.3	19.8	69.7%	12.1%	[42]
First-line	Pilot	Single arm	CBDCA + S-1	21	4.2	9.7	33.0%	9.5%	[43]
First-line	2	Single arm	CBDCA + S-1	33	4.8	12.8	33.3%	6.1%	[44]
First-line	2	Single arm	CBDCA + weekly PTX + Bev	17	5.7	12.9	52.9%	5.9%	[45]
First-line	3	Randomized control trial	CBDCA + nab-PTX	120	5.5	13.0	56.0%	1.6%	[46]
			CBDCA + nab-PTX + Nintedanib	120	6.2	15.3	69.0%	4.1%	
Second-line	Pilot	Single arm	Nivolumab	6			50%	0.0%	[33]
Second-line	2	Single arm	Nivolumab	18	7.4	15.6	39%	11.1%	[34]
Second-line	2	Single arm	Atezolizumab	17 (Stopped)	3.4		6.3%	29.4%	[35]

Abbreviations: PFS, progression-free survival; OS, overall survival; ORR, overall response rate; AE, acute exacerbation; IP, interstitial pneumonia; CBDCA, carboplatin; nab-PTX, nanoparticle albumin-bound paclitaxel; PTX, paclitaxel; Bev, bevacizumab.

**Table 2 cancers-16-01734-t002:** Drug interactions between molecularly targeted drugs and rifampicin.

Main Target	Drugs	Metabolic Mediator	Decrease of AUC	Decrease of Cmax
EGFR	Gefitinib	CYP3A4	83%	65%
	Erlotinib	CYP3A4	69%	39%
	Afatinib	P-glycoprotein	34%	22%
	Dacomitinib	CYP2D6	No data	No data
	Osimertinib	CYP3A	78%	73%
ALK	Alectinib	CYP3A4	73%	51%
	Lorlatinib	CYP3A, UGT1A4	85%	76%
	Brigatinib	CYP2C8, CYP3A4	80%	60%
	Ceritinib	CYP2C9, CYP3A4	70%	44%
ROS-1	Crizotinib	CYP3A4	84%	79%
ROS-1/NTRK	Entrectinib	CYP3A	77%	55%
BRAF	Dabrafenib	CYP2C8/9, CYP3A4	34%	27%
	Trametinib	CYP2B6, CYP3A4	No data	No data
MET	Tepotinib	CYP2C8/9, CYP3A4	No data	No data
	Capmatinib	CYP3A4	67%	56%
RET	Selpercatinib	CYP3A4	87%	70%
KRAS	Sotorasib	CYP3A	51%	35%
HER2	Trastuzumab	CYP3A	No data	No data

Abbreviations: AUC, area under the concentration–time curve; Cmax, maximum plasma concentration; EGFR, epidermal growth factor receptor; ALK, anaplastic lymphoma kinase; ROS-1, c-ros oncogene 1; NTRK, neurotrophic receptor tyrosine kinase; BRAF, B-Raf proto-oncogene, serine/threonine kinase; MET, mesenchymal-epithelial transition; RET, rearranged during transfection; KRAS, v-Ki-ras2 Kirsten rat sarcoma viral oncogene homolog; HER2, human epidermal growth factor type 2: CYP, cytochrome P450; UGT, UDP-glucuronosyl transferase.

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
