# Peer review of "Treatment Strategies for Non-Small-Cell Lung Cancer with Comorbid Respiratory Disease; Interstitial Pneumonia, Chronic Obstructive Pulmonary Disease, and Tuberculosis"

_cancers, 2024, doi:10.3390/cancers16091734_

Round 1
Reviewer 1 Report
Comments and Suggestions for Authors
This review centered on managing non-small cell lung cancer (NSCLC) complicated by interstitial pneumonia (IP), chronic obstructive pulmonary disease (COPD), and pulmonary tuberculosis (TB). The authors suggest carboplatin plus nanoparticle albumin-bound paclitaxel as a safe first-line treatment for NSCLC with IP. Immune checkpoint inhibitors (ICIs) show potential benefits despite safety concerns. COPD severity influences prognosis, and TB risk increases with pharmacotherapy, particularly ICIs. The review discusses treatment challenges and future strategies. The manuscript required major modifications in its present form.
1. The title of this review is not so attractive as it is currently too wordy. Consider revising it to better match the focus of the review.
2. It is important to note that many cited articles are over a decade old, I recommend incorporating more recent research for a comprehensive understanding.
3. Highlight the novelty of this review compared to other available literature in the abstract section.
4. Recently, nanosecond pulses of the electromagnetic field were used to successfully treat NSCLC [https://doi.org/10.1016/j.fmre.2024.02.001]. For the broad range readership of this review, I recommend incorporating these interesting findings in this review as the latest information which might also have the potential to deal with lung carcinoma (NSCLC).
4. The immunological mechanisms that contribute to the development of fatal acute exacerbation in NSCLC patients with (IP undergoing pharmacotherapy should be explained in detail.
5. What are the molecular pathways linking COPD severity with prognosis in NSCLC patients, and how might these pathways influence drug selection and treatment outcomes? Any study available on this subject is encouraged to discuss in this review.
6. In NSCLC patients with IP, how does nanoparticle albumin-bound paclitaxel interact with the pulmonary microenvironment, and what qualities make it a safer option than conventional paclitaxel formulations? Explain in detail.
7. The conclusion should convey the key insights derived from the literature review in the manuscript, providing clear and key information for the reader. Unfortunately, currently, none of these are available in this section. Rewrite the whole conclusion section.
6. Add perspective and future challenges after or in conclusion which is currently missing.
7. The addition of graphical figures would enhance the concept and readability of the review article. Authors are encouraged to consider using visual representations to illustrate key concepts and findings discussed in the text. (optional).
8. I recommend thoroughly reviewing the manuscript to identify and rectify any typos and grammatical errors.
Comments on the Quality of English LanguageI recommend thoroughly reviewing the manuscript to identify and rectify any typos and grammatical errors.
Author Response
Reviewer 1's comments:
This review centered on managing non-small cell lung cancer (NSCLC) complicated by interstitial pneumonia (IP), chronic obstructive pulmonary disease (COPD), and pulmonary tuberculosis (TB). The authors suggest carboplatin plus nanoparticle albumin-bound paclitaxel as a safe first-line treatment for NSCLC with IP. Immune checkpoint inhibitors (ICIs) show potential benefits despite safety concerns. COPD severity influences prognosis, and TB risk increases with pharmacotherapy, particularly ICIs. The review discusses treatment challenges and future strategies. The manuscript required major modifications in its present form.
<Response>
Thank you for your encouraging comment. We have carefully read and addressed all your comments.
- The title of this review is not so attractive as it is currently too wordy. Consider revising it to better match the focus of the review
<Response>
Thank you for your comment. As you indicated, we tried to shorten the title, but could not shorten it any further. Therefore, we did not change the title this time.
- It is important to note that many cited articles are over a decade old, I recommend incorporating more recent research for a comprehensive understanding.
<Response>
Thank you for your comment. As you have pointed out, about one-third of the references in our paper are more than 10 years old. We know that it is preferable to refer to newer studies, but we hope you understand that there are few studies on the subject of lung cancer with comorbid other respiratory diseases. As you indicated, we have added some relatively new papers as below.
References
- Nakajima M, Yamamoto N, Hayashi K, Karube M, Ebner DK, Takahashi W, Anzai M, Tsushima K, Tada Y, Tatsumi K, Miyamoto T, Tsuji H, Fujisawa T, Kamada T. Carbon-ion radiotherapy for non-small cell lung cancer with interstitial lung disease: a retrospective analysis. Radiat Oncol. 2017;12:144.
- Juie Nahushkumar Rana, Sohail Mumtaz, Ihn Han, Eun Ha Choi. Formation of reactive species via high power microwave induced DNA damage and promoted intrinsic path-way-mediated apoptosis in lung cancer cells: An in vitro investigation. Fundamental Research. 2024 (in press). https://doi.org/10.1016/j.fmre.2024.02.001.
- Delaunois LM. Mechanisms in pulmonary toxicology. Clin Chest Med. 2004;25:1-14.
- Isono T, Kagiyama N, Takano K, Hosoda C, Nishida T, Kawate E, Kobayashi Y, Ishiguro T, Takaku Y, Kurashima K, Yanagisawa T, Takayanagi N. Outcome and risk factor of immune-related adverse events and pneumonitis in patients with advanced or postoperative recurrent non-small cell lung cancer treated with immune checkpoint inhibitors. Thorac Cancer. 2021;12:153-164.
- Tasaka Y, Honda T, Nishiyama N, Tsutsui T, Saito H, Watabe H, Shimaya K, Mochizuki A, Tsuyuki S, Kawahara T, Sakakibara R, Mitsumura T, Okamoto T, Kobayashi M, Chiaki T, Yamashita T, Tsukada Y, Taki R, Jin Y, Sakashita H, Natsume I, Saitou K, Miyashita Y, Miyazaki Y. Non-inferior clinical outcomes of immune checkpoint inhibitors in non-small cell lung cancer patients with interstitial lung disease. Lung Cancer. 2021;155:120-126.
- Zhou J, Chao Y, Yao D, Ding N, Li J, Gao L, Zhang Y, Xu X, Zhou J, Halmos B, Tsoukalas N, Kataoka Y, de Mello RA, Song Y, Hu J. Impact of chronic obstructive pulmonary disease on immune checkpoint inhibitor efficacy in advanced lung cancer and the potential prognostic factors. Transl Lung Cancer Res. 2021;10:2148-2162.
- Shin SH, Park HY, Im Y, Jung HA, Sun JM, Ahn JS, Ahn MJ, Park K, Lee HY, Lee SH. Improved treatment outcome of pembrolizumab in patients with nonsmall cell lung cancer and chronic obstructive pulmonary disease. Int J Cancer. 2019;145:2433-2439.
- Highlight the novelty of this review compared to other available literature in the abstract section
<Response>
Thank you for your valuable comment. Based on your comment, we added the sentence as below.
Page 1, line 36
Since pharmacotherapy, especially ICIs, reportedly induces the development of TB, the possibility of developing TB should always be kept in mind during NSCLC treatment. To date, there is no coherent review article on NSCLC with these pulmonary complications. This review article summarizes the current evidence and discusses future prospects for treatment strategies for NSCLC patients complicated with IP, severe COPD, and TB.
- Recently, nanosecond pulses of the electromagnetic field were used to successfully treat NSCLC [https://doi.org/10.1016/j.fmre.2024.02.001]. For the broad range readership of this review, I recommend incorporating these interesting findings in this review as the latest information which might also have the potential to deal with lung carcinoma (NSCLC).
<Response>
Thank you for your valuable comment. Based on your comment, we added the sentence as below.
Page 2, line 75
Recently, LC treatment has made significant advances, and newer therapies such as carbon-ion radiotherapy and high-power microwave expected to be relatively safe treatment options for LC patients with IP complications [20,21]. However, most LC patients are already advanced at diagnosis, and pharmacotherapy remains the cornerstone of treatment. In particular, clinicians are often faced with the difficult choice of pharmacotherapy for patients with non-small cell lung carci-noma (NSCLC), which has various treatment options, including molecular-targeted drugs and immune checkpoint inhibitors (ICIs) in addition to cytotoxic chemotherapy.
- The immunological mechanisms that contribute to the development of fatal acute exacerbation in NSCLC patients with (IP undergoing pharmacotherapy should be explained in detail.
<Response>
Thank you for your comment. We believe that the point you note is very important, but the exact mechanism of chemotherapy-induced acute exacerbation of interstitial pneumonia is not yet known. Therefore, we have added the following information on the commonly proposed mechanism.
Page 2, line 88
Based on previous reports, pharmacotherapy for NSCLC patients with comorbid IP induces acute exacerbation of IP with a frequency of approximately 5–20% [17-19]. The exact mechanism by which pharmacotherapy causes acute exacerbations of IP is not known, but direct cell damage by reactive oxygen species and proteolytic enzymes, and activation of immune cells have been proposed [22]. The risk factors for the development of acute exacerbations of IP due to cytotoxic chemotherapy have been reported in several studies.
- What are the molecular pathways linking COPD severity with prognosis in NSCLC patients, and how might these pathways influence drug selection and treatment outcomes? Any study available on this subject is encouraged to discuss in this review.
<Response>
Thank you for your comment. In the previous studies of lung cancer complicated by COPD, no molecular pathway has been identified that associate COPD severity and lung cancer prognosis. Therefore, we have added the following sentence based on your comment.
Page 7, line 288
Furthermore, the severity of COPD is considered an important prognostic factor in LC patients, and it has been reported that the more severe the Global Initiative for Chronic Obstructive Lung Disease classification grade of COPD severity, the worse the prognosis of LC patients [58]. No molecular pathway linking COPD severity and LC prognosis has been identified, and elucidation of this molecular pathway may contribute to treatment selection for LC patients with severe COPD. LC patients with severe COPD have not only low pulmonary function but also a high frequency of systemic complications, including cancer cachexia, heart failure, and diabetes mellitus, and many patients have a poor performance status (PS).
- In NSCLC patients with IP, how does nanoparticle albumin-bound paclitaxel interact with the pulmonary microenvironment, and what qualities make it a safer option than conventional paclitaxel formulations? Explain in detail.
<Response>
Thank you for your comment. Since no direct comparison study between paclitaxel and nab-paclitaxel has been reported to date, it is not clear which is safer. Previous reports have described that nab-paclitaxel acts based on the enhanced permeability and retention (EPR) effect, and exhibits lower toxicity than non-EPR-based agents owing to its low perfusion into organs like the kidney and heart [Cancer Sci. 2013;104:779–789, Biomaterials. 2017;140:162–169]. However, there are no reports on the mechanism by which nab-PTX has less impact on IP. We also asked Taiho Pharmaceutical, the Japanese distributor of nab-PTX, but they replied that the mechanism was unknown. Therefore, we could not add a reason why nab-PTX is less likely to cause acute exacerbations of IP at this time.
- The conclusion should convey the key insights derived from the literature review in the manuscript, providing clear and key information for the reader. Unfortunately, currently, none of these are available in this section. Rewrite the whole conclusion section.
<Response>
Thank you for your valuable comment. As you indicated, we added the sentence as below.
Page 12, line 531
LC patients are often complicated by other respiratory diseases, with the manage-ment of which being problematic. This review article summarizes the current evidence and discusses future prospects for treatment strategies for LC patients complicated by IP, severe COPD, and TB.
Approximately 5–15% of LC patients have IP, and it is most important to select a treatment that is less likely to induce acute exacerbations of pre-existing IP. For first-line treatment of advanced NSCLC with comorbid IP, carboplatin plus nab-PTX is the treat-ment regimens with the most reported efficacy and safety. Although the safety of ICIs for NSCLC with IP is still controversial, it has the potential to be the only existing therapy with long-term survival.
The prevalence of COPD in LC patients is approximately 40–70%, the severity of COPD is considered an important prognostic factor in LC patients. However, even in the cases of low pulmonary function requiring home oxygen therapy, up to the first-line or second-line treatment may be considered. In pharmacotherapy for NSCLC with severe COPD, it is important to select agents that cause fewer pulmonary adverse events and cardiovascular burden. In addition, clinicians should not forget to treat COPD.
Active TB is noted in 2–5% of LC cases, whereas LC is noted in 1–2% of active TB cases. When treating LC patients with pharmacotherapy, especially ICIs, the risk of developing TB should always be kept in mind. Ideally, it may be desirable to precede TB treatment until the risk of TB infection has decreased, but this should not result in the loss of the opportunity for radical treatment of LC. In the treatment of LC patients with active TB, consider adding LC treatment after 2-3 weeks of prior TB treatment.
- Add perspective and future challenges after or in conclusion which is currently missing.
<Response>
Thank you for your comment. Based on your comment, we added the sentence as below.
Page 12, line 554
In particular, although the safety of ICIs for NSCLC with these complications is still controversial, it has the potential for long-term survival. It is particularly important to identify risk factors and biomarkers that predict exacerbation of preexisting lung disease by ICIs. For appropriate patient selection, large studies are warranted in the future to identify the risk factors for ICIs in NSCLC patients with these complications.
- The addition of graphical figures would enhance the concept and readability of the review article. Authors are encouraged to consider using visual representations to illustrate key concepts and findings discussed in the text. (optional).
<Response>
Thank you for your valuable comment. We believe that the point you note is very important. In lung cancer patients with IP, COPD, and Tb, the problem in clinical practice is that pharmacotherapy for lung cancer can exacerbates these underlying diseases. On the other hand, because lung cancer is the most directly related to their prognosis, clinicians must treat lung cancer with the risk of deterioration of the underlying disease. We attempted to represent this dilemma in a schematic, but the specificity of each disease and lack of evidence made it difficult to represent it in a single diagram. Therefore, we hope you understand that we were not able to create an additional figure at this time.
- I recommend thoroughly reviewing the manuscript to identify and rectify any typos and grammatical errors.
<Response>
Thank you for your comment. Although this paper has been proofread in English, as you indicated, I read it again to check for grammatical errors and typos.
Reviewer 2 Report
Comments and Suggestions for Authors
This is a very throughout review discuss the treatment methods for NSCLC with ILD, COPD and TB co-existence. The 2 tables provided in the review showed the most current methods for treatment when encountering the complication of multiple diseases.
If the author can provide a schematics to show the difficulties of why the treatments currently are not able to treat both the diseases in the same time, that will be much helpful to understand the disadvantage of current research and what kind of direction should be going later.
Also, for this review, it looks like a bit discontinued, with 3 topics, if the authors can provide a overall introduction and then discuss 3 different aspects, that will help to first understand the co-existence of lung disease with NSCLC is difficult in treatment while then go to in details.
One minor suggestion, if any table can be provided for COPD with NSCLC treatment ?
Author Response
Reviewer 2's comments:
This is a very throughout review discuss the treatment methods for NSCLC with ILD, COPD and TB co-existence. The 2 tables provided in the review showed the most current methods for treatment when encountering the complication of multiple diseases.
<Response>
Thank you for your encouraging comment. We have carefully read and addressed all your comments.
If the author can provide a schematics to show the difficulties of why the treatments currently are not able to treat both the diseases in the same time, that will be much helpful to understand the disadvantage of current research and what kind of direction should be going later.
<Response>
Thank you for your valuable comment. We believe that the point you note is very important. In lung cancer patients with IP, COPD, and Tb, the problem in clinical practice is that pharmacotherapy for lung cancer can exacerbates these underlying diseases. On the other hand, because lung cancer is the most directly related to their prognosis, clinicians must treat lung cancer with the risk of deterioration of the underlying disease. We attempted to represent this dilemma in a schematic you pointed out, but the specificity of each disease and lack of evidence made it difficult to represent it in a single diagram. Therefore, we hope you understand that we were not able to create an additional figure at this time.
Also, for this review, it looks like a bit discontinued, with 3 topics, if the authors can provide a overall introduction and then discuss 3 different aspects, that will help to first understand the co-existence of lung disease with NSCLC is difficult in treatment while then go to in details.
<Response>
Thank you for your comment. Despite your kind advice, the structure of this review has not allowed us to prepare an introduction summarizing the three diseases. We hope you will find the brief background and purpose of this review article in the “Simple Summary” and “Abstract”.
One minor suggestion, if any table can be provided for COPD with NSCLC treatment ?
<Response>
Thank you for your comment. We were able to create table 1 and summarize the studies on lung cancer complicated by IP because several phase II and phase III trials exist. However, there are no interventional studies on lung cancer complicated by severe COPD and evidence is lacking. Personally, I believe that carboplatin plus nab-PTX or S1 monotherapy is the appropriate treatment choice for these populations as well as IP, but due to the lack of evidence, I did not include it in the table this time.
Reviewer 3 Report
Comments and Suggestions for Authors
1. Line 258, 293, 306, 344 and 352 - whenever mentioned the word “significant” – should also reporting the corresponding p-value to support such statement.
2. Line 480 – based on “previous reports” to generate Figure 1 – should list the citations/name & source of these reports.
3. Suggested to add a section mention the limitation of the method/study.
Author Response
Reviewer 3's comments:
- Line 258, 293, 306, 344 and 352 - whenever mentioned the word “significant” – should also reporting the corresponding p-value to support such statement.
<Response>
Thank you for your comment. As you pointed out, the p-values should have been listed in all places where they were expressed as “significant”. Based on your comment, we added or changed the sentence as below. In the section of the article on the effect of pirfenidone in reducing postoperative acute exacerbations, the word “significantly” was removed because this is not a placebo-controlled study.
Page 2, line 61
In an observational study involving 181 IPF patients, Tomassetti et al. reported that the median survival of IPF patients with LC was significantly shorter than that of IPF patients without LC (38.7 vs 63.9 months, p < 0.001) [10].
Page 6, line 267
Furthermore, in a multicenter phase II trial of perioperative treatment of NSCLC with IPF, pirfenidone significantly reduced the incidence of postoperative acute exacerbations of IPF [50].
Page 7, line 306
The Japanese retrospective study evaluating the efficacy and safety of chemotherapy in 40 patients with advanced LC complicated by chronic respiratory failure requiring home oxygen therapy also showed that the only factor significantly associated with improved prognosis was the use of first-line or second-line treatment (HR, 0.42; 95% CI: 0.18–0.94) [61].
Page 7, line 319
LC patients with severe or most severe COPD (%FEV1 < 50%) reported a significantly higher rate of pulmonary adverse events such as pneumonia (46.4% vs 31.2%, p < 0.001) and COPD exacerbations (30.4% vs 6.9%, p < 0.001) during LC treatment than those with mild-to-moderate COPD (%FEV1 > 50%) [62].
Page 8, line 360
Additionally, a subset analysis of this study reported significantly longer survival in ex-smokers with COPD complications than in ex-smokers without COPD (OS; 359 vs 145 days, p = 0.035).
- Line 480 – based on “previous reports” to generate Figure 1 – should list the citations/name & source of these reports.
<Response>
Thank you for your comment. As you pointed out, we added the citations of these reports.
Page 10, line 495
In our opinion, we offer the following treatment suggestions based on the previous re-ports [92,103,104,113,114] (Figure 1).
- Suggested to add a section mention the limitation of the method/study.
<Response>
Thank you for your valuable comment. Because our paper is a review article, we have added the following sentence to the conclusion instead of listing it as a limitation.
Page 12, line 550
This review article summarizes the current evidence and discusses future prospects for treatment strategies for LC patients complicated by IP, severe COPD, and TB. However, large prospective studies on LC patients complicating these respiratory diseases are limited and the evidence is insufficient. In particular, although the safety of ICIs for NSCLC with these complications is still controversial, it has the potential for long-term survival.
Round 2
Reviewer 1 Report
Comments and Suggestions for Authors
I completely agree with the revised version. I recommend accepting the paper for publication in its present form.